# *Rosa davurica* Pall., *Rosa rugosa* Thumb., and *Rosa acicularis* Lindl. Originating from Far Eastern Russia: Screening of 146 Chemical Constituents in Three Species of the Genus *Rosa*

Mayya P. Razgonova [1,2,*], Bayana A. Bazhenova [3], Yulia Yu. Zabalueva [4], Anastasia G. Burkhanova [3], Alexander M. Zakharenko [5,6], Andrey N. Kupriyanov [7], Andrey S. Sabitov [1], Sezai Ercisli [8] and Kirill S. Golokhvast [2,5,6]

1 N.I. Vavilov All-Russian Institute of Plant Genetic Resources, B. Morskaya 42-44, 190000 Saint-Petersburg, Russia
2 Department of Bioeconomy and Food Security, Far Eastern Federal University, Sukhanova 8, 690950 Vladivostok, Russia
3 East Siberian State University of Technology and Management, Klyuchevskaya Str. 40V, 670013 Ulan-Ude, Russia
4 K.G. Razumovsky Moscow State University of Technologies and Management, Zemlyanoy Val Str. 73, 109004 Moscow, Russia
5 Siberian Federal Scientific Centre of Agrobiotechnology, Centralnaya, Presidium, 633501 Krasnoobsk, Russia
6 Laboratory of Supercritical Fluid Research and Application in Agrobiotechnology, Tomsk State University, Lenin Str. 36, 634050 Tomsk, Russia
7 Federal Research Center of Coal and Coal-Chemistry of SB RAS, 650000 Kemerovo, Russia
8 Ataturk University, Kampusu Ataturk Universitesi, 25030 Yakutiye, Turkey
* Correspondence: m.razgonova@vir.nw.ru

**Abstract:** *Rosa rugosa* Thumb., *Rosa davurica* Pall., and *Rosa acicularis* Lindl. contain a large number of target analytes which are bioactive compounds. High performance liquid chromatography (HPLC), in combination with the ion trap (tandem mass spectrometry), was used to identify target analytes in MeOH extracts of *R. rugosa*, *R. davurica*, and *R. acicularis*, originating from the Russian Far East, Trans-Baikal Region, and Western Siberia. The results of initial studies revealed the presence of 146 compounds, of which 115 were identified for the first time in the genus *Rosa* (family *Rosaceae*). The newly identified metabolites belonged to 18 classes, including 14 phenolic acids and their conjugates, 18 flavones, 7 flavonols, 2 flavan-3-ols, 2 flavanones, 3 stilbenes, 2 coumarins, 2 lignans, 9 anthocyanins, 3 tannins, 8 terpenoids, 3 sceletium alkaloids, 4 fatty acids, 2 sterols, 2 carotenoids, 3 oxylipins, 3 amino acids, 5 carboxylic acids, etc. The proven richness of the bioactive components of targeted extracts of *R. rugosa*, *R. davurica*, and *R. acicularis* invites extensive biotechnological and pharmaceutical research, which can make a significant contribution both in the field of functional and enriched nutrition, and in the field of cosmetology and pharmacy.

**Keywords:** *Rosa rugosa*; *Rosa davurica*; *Rosa acicularis*; ion trap; tandem mass spectrometry; polyphenolic compounds

## 1. Introduction

Plants have been used as medicines since the existence of human civilization [1,2]. More than 35 thousand varieties of plants from different parts of the world are actively used for medical purposes, since they contain numerous phytocomponents that can potentially treat many diseases, including infectious ones [3]. Numerous medical systems of treatment, such as Ayurveda, Unani, homeopathy, naturopathy, Siddha, and others, rely on plants as effective remedies for various life-threatening diseases [4,5]. Due to the presence of secondary metabolites in plants, they have significant potential as antimicrobial agents. The diversity of these natural products offers an endless number of possibilities for the discovery of new drugs for the treatment of various diseases [6–8].

In recent years, traditional medicine based on oral herbal preparations has attracted the attention of both consumers and healthcare professionals. However, the use of these medicinal products requires improved knowledge of their composition and stability over time in order to support or validate these therapies in humans. Liquid preparations from medicinal plants, such as tinctures and extracts from plant buds, are typical products that are widely used but still poorly understood. Plant bud extracts are defined as extracts obtained exclusively from fresh buds, shoots, young leaves, and/or roots, which are macerated and extracted with hydro–glycerol and water–alcohol mixtures [9]. Kidney extracts represent a new category of herbal products well known and widely used in gemmotherapy, as well as in homeopathy and herbal medicine [10].

The genus *Rosa* (family *Rosaceae*) is represented on the territory of the Trans-Baikal region, Far East (Russian Federation), and Western Siberia by 3 species—*Rosa rugosa* Thumb., *Rosa davurica* Pall., and *Rosa acicularis* Lindl. (Figures 1 and 2). Fresh fruits and leaves contain up to 900 mg% ascorbic acid per dry pulp weight. Fresh petals contain 0.25–0.38% essential oil. Its neutral volatile fraction contains 86.3% phenylethyl alcohol, some linalool, citronellol, geraniol, nerol, etc. Eugenol was found in the phenolic fraction, phenylacetic, benzoic, and other acids in the acid fraction. *R. rugosa* is a medicinal plant widely used in traditional and folk medicine. Extracts of *R. rugosa* have been valued for Asian culinary, cosmetic, and aromatherapy purposes, and used in herbal medicines for diabetes mellitus and osteoarthritis [11]. The medicinal effects seem to be involved in the presence of many phytochemicals in *R. rugosa* extracts, for example flavonoids, phenylpropanoid, tannins, fatty acids, and terpenoids [12].

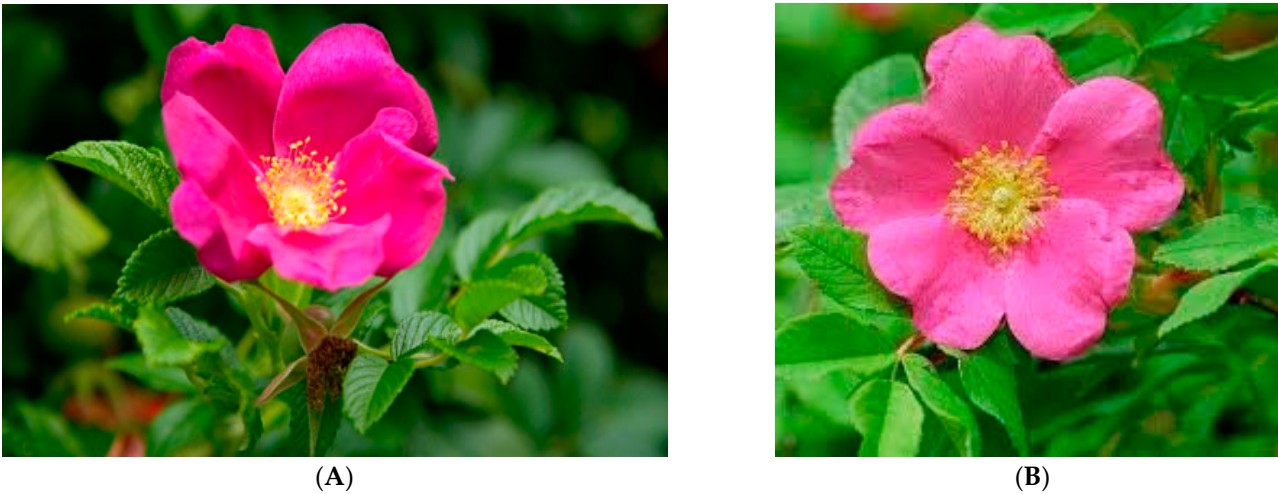

| (**A**) | (**B**) |

**Figure 1.** (**A**) *Rosa rugosa* (Far Eastern Russia); (**B**) *Rosa davurica* (Trans-Baikal region).

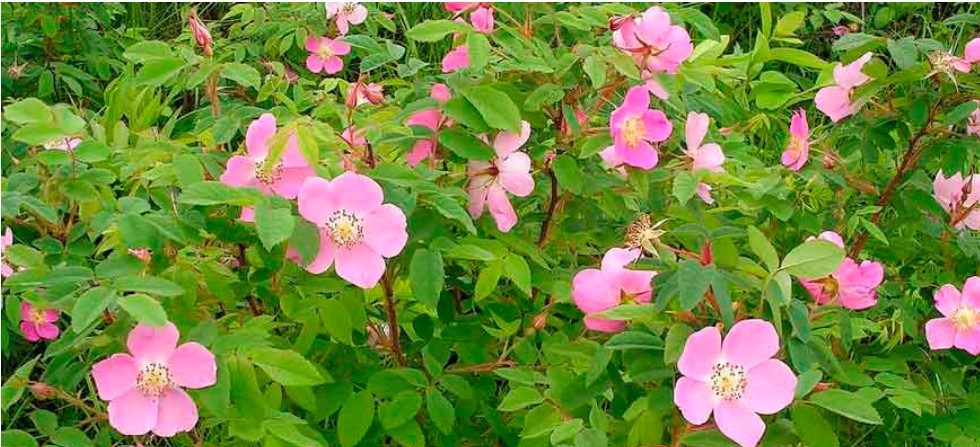

**Figure 2.** *Rosa acicularis* (Western Siberia).

Several studies have reported that some compounds from rose hip extracts exhibit anti-inflammatory activity in vitro. The anti-inflammatory property of the crude hydroalcoholic extract of rose hip has been proven in vivo, suggesting its potential role as one of the main therapies for the treatment of diseases associated with inflammation [13]. In Turkish folk medicine, a decoction of fresh rose hips is prepared and used to treat various stomach disorders [14]. Trans-Tiliroside (Tribuloside) has been found to be one of the main active components of aqueous acetone extracts from fruits and seeds that inhibit weight gain and lower plasma triglyceride levels in mice [15]. Additionally, clinical studies have demonstrated the positive effect of rose hip powder in the treatment of osteoarthritis [16]. Rose hip powder enhances in vitro anti-inflammatory and chondroprotective properties in leukocytes and primary chondrocytes of human peripheral blood [17]. Unfortunately, to date, there are few data providing information on the biological action of extracts of buds and leaves, and it should be noted that these preparations have never been used for preclinical and clinical trials.

The present investigation was designed to carry out a phytochemical study involving detailed metabolomic and comparative analysis of fruits and flowers of *Rosa rugosa* Thumb., *Rosa davurica* Pall., and *Rosa acicularis* Lindl. originating from the Trans-Baikal region, Western Siberia, and Russian Far East.

## 2. Results

Approximately 300 mass spectra were assessed for each analytical replicate and MS operating condition in this comprehensive approach for a complete screening of phytochemicals (Figure 3).

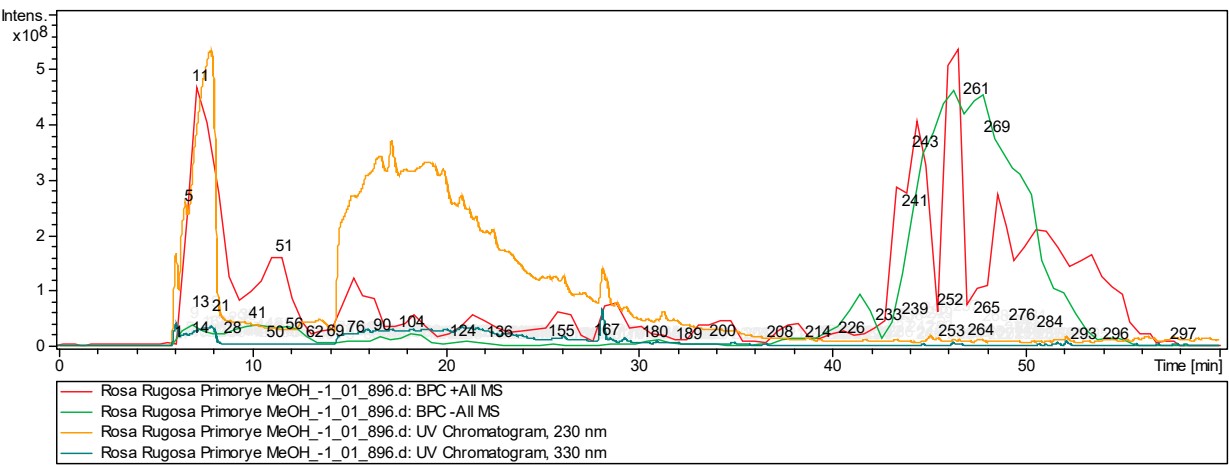

**Figure 3.** Representative chemical profiles of the *R. rugosa* (Primorye, Russia) total ion chromatogram from the MeOH extract.

This procedure allowed a detailed evaluation of the rose MeOH extract fraction and the tentative identification of up to 146 phytochemicals (Table A1 (Appendix A)). The most represented classes of polyphenolic compounds were flavonoids (flavonols, flavones, flavan-3-ols, flavanones) with a total of 68 polyphenols identified for the first time. Some polyphenols were identified for the first time in the genus *Rosa* (family *Rosaceae*).

These are the flavones: Chrysoeriol, Hispidulin, 5,7-Dimethoxyluteolin, Cirsimaritin, Cirsiliol, Tricin, Jaceosidin, Nevadensin, Syringetin, Isovitexin, Genistein *C*-glucoside malonylated, Chrysin 6-*C*-glucoside-6''-*O*-deoxyhexoside; flavanols: Dihydrokaempferol, Rhamnetin II, Kaempferol-3-*O*-α-L-rhamnoside, Taxifolin-*O*-pentoside, Taxifolin-3-*O*-hexoside, Isorhamnetin triacetyl hexoside; flavan-3-ols: Epiafzelechin and Gallocatechin; flavanone: Naringenin, Fustin; phenolic acids: Caffeic acid, Citric acid, Hydroxy methoxy dimethylbenzoic acid, Hydroxyferulic acid, Ellagic acid, *p*-Coumaroylquinic acid, Ginkgoic acid, Salvianolic acid D, Salvianolic acid B; stilbenes: Pinosylvin, Resveratrol, 3-Hydroxyresveratrol;

lignans: Pinoresinol, Arctigenin; coumarins: 3,4,5–Trimethoxycoumarin, Fraxin; anthocyanins: Cyanidin 3-*O*-glucoside, Delphinidin *O*-pentoside, Pelargonidin 3-*O*-(6-*O*-malonyl-β-D-glucoside), Cyanidin 3-(6″-Succinyl-Glucoside), Delphinidin malonyl hexoside, Cyanidin 3-*O*-dioxayl-glucoside, Delphinidin 3,5-dihexoside, etc.

## 3. Discussion

A total of 146 compounds were identified in extracts of *Rosa rugosa* Thumb., *Rosa davurica* Pall., and *Rosa acicularis* Lindl. based on their accurate MS, fragment ions, and by searching online databases and the reported literature. A total of 115 compounds were identified for the first time in the genus *Rosa* (family *Rosaceae*). The newly identified metabolites belonged to 18 classes, including 14 phenolic acids and their conjugates, 18 flavones, 7 flavonols, 2 flavan-3-ols, 2 flavanones, 3 stilbenes, 2 coumarins, 2 lignans, 9 anthocyanins, 3 tannins, 8 terpenoids, 3 sceletium alkaloids, 4 fatty acids, 2 sterols, 2 carotenoids, 3 oxylipins, 3 amino acids, 5 carboxylic acids, etc. Metabolomic screening of polyphenols from extracts of *R. rugosa*, *R. davurica*, and *R. acicularis* included flavones, flavonols, flavan-3-oles, flavanones, anthocyanins, condensed tannins, phenolic acids, lignans, stilbenes, and coumarins.

### 3.1. Dimethoxyflavones

The flavones 5,7-Dimethoxyluteolin (compound **5**), Cirsimaritin (compound **6**), Chrysoeriol methyl ether (compound **7**), Cirsiliol (compound **8**), Tricin (compound **9**), Jaceosidin (compound **10**), and Syringetin (compound **13**) (Table A1 (Appendix A)) have been already characterized as components of *Syzygium aromaticum* [18], *Ocimum* [19], *Rosmarinus officinalis* [20], Bougainvillea [21], *Triticum aestivum* [22]; millet grains [23]; *Sasa veitchii; Phyllostachys nigra* [24], etc. Thus, the flavone Jaceosidin was found in extracts of *R. davurica*. The flavone 5,7-Dimethoxyluteolin was found in extracts of *R. rugosa* and *R. davurica*. The CID-spectrum (collision induced dissociation spectrum) in negative ion modes of Tricin from extracts of *R. davurica* is shown in Figure 4.

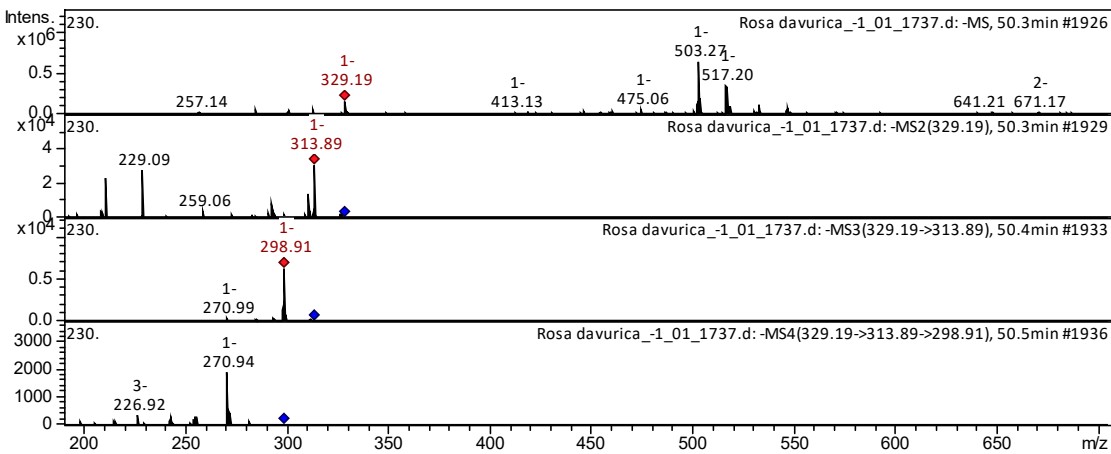

**Figure 4.** CID-spectrum of Tricin from extracts of *R. davurica*, *m/z* 329.19.

The [M – H]⁻ ion produced three fragment ions at *m/z* 313, *m/z* 259, and *m/z* 229 (Figure 4). The fragment ion with *m/z* 313 produced two daughter ions at *m/z* 298 and *m/z* 271. The fragment ion with *m/z* 298 yielded two daughter ions at *m/z* 271 and *m/z* 227. It was identified in the bibliography in extracts of *Triticum aestivum* [22]; millet grains [23]; *Sasa veitchii; Phyllostachys nigra* [24].

### 3.2. Trimethoxyflavones

The flavones Nevadensin (compound **12**) and Pentahydroxy trimethoxy flavone (compound **15**) (Table A1 (Appendix A)) have been already characterized as components of *Ocimum* [19], *F. glaucescens; C. edulis* [25], *Mentha* [26], etc. Thus, the flavone Nevadensin was

found in extracts of *R. acicularis.* The CID-spectrum in positive ion modes of Nevadensin from extracts of *R. acicularis* is shown in Figure 5.

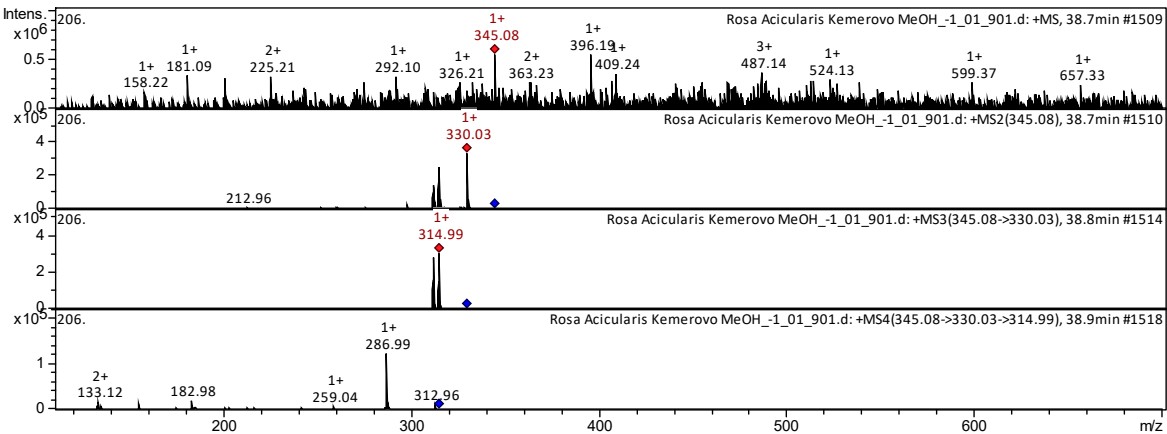

**Figure 5.** CID-spectrum of Nevadensin from extracts of *R. acicularis*, *m/z* 346.86.

The [M + H]⁺ ion produced two fragment ions at *m/z* 330 and *m/z* 212 (Figure 5). The fragment ion with *m/z* 330 yielded one daughter ion at *m/z* 314. The fragment ion with *m/z* 314 yielded five daughter ions at *m/z* 312, *m/z* 286, *m/z* 259, *m/z* 182, and *m/z* 133. It was identified in the bibliography in extracts of *Ocimum* [19] and *Mentha* [26].

### 3.3. Trihydroxyflavones

The flavones Apigenin (compound **2**), Chrysoeriol (compound **3**), Isovitexin (compound **16**), and flavonol Isokaempferide (compound **22**) have been already characterized as components of *Mentha* [26], *Hedyotis diffusa* [27], *Andean blueberry* [28], *Stevia rebaudiana* [29], *Rosa rugosa* [30], Propolis [31], *Rhus coriaria* [32], Mexican lupine species [33], etc. Thus, the flavonol Isokaempferide was found in extracts of *R. davurica.* The CID-spectrum in positive ion modes of Isokaempferide from extracts of *R. davurica* is shown in Figure 6.

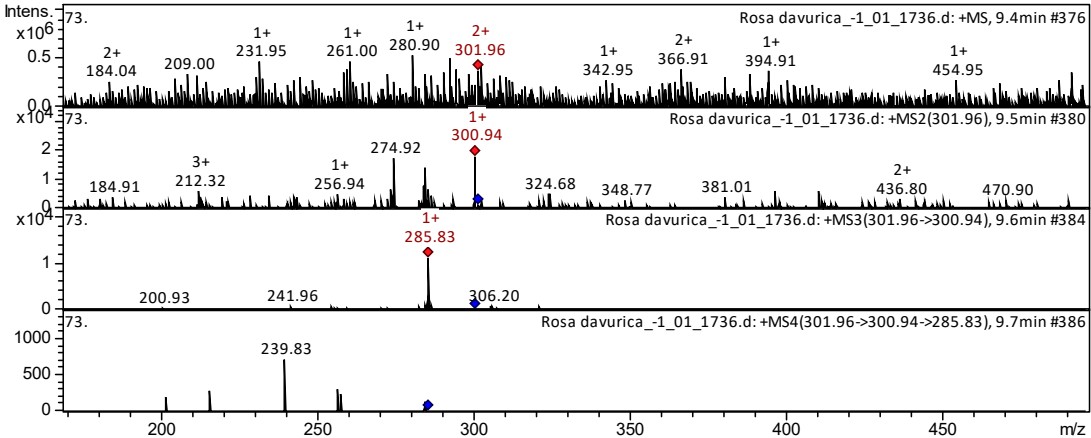

**Figure 6.** CID-spectrum of Isokaempferide from extracts of *R. davurica*, *m/z* 301.96.

The [M + H]⁺ ion produced five fragment ions at *m/z* 300, *m/z* 274, *m/z* 256, *m/z* 212, and *m/z* 184 (Figure 6). The fragment ion with *m/z* 300 yielded three daughter ions at *m/z* 285, *m/z* 241, and *m/z* 200. The fragment ion with *m/z* 285 yielded one daughter ion at *m/z* 239. It was identified in the bibliography in extracts of *Rosa rugosa* [30] and *Propolis* [31].

### 3.4. Tetrahydroxyflavones

The flavonols Kaempferol (compound **20**), Dihydrokaempferol (compound **21**), Kaempferol-3-*O*-α-ʟ-rhamnoside (compound **30**), Kaempferol diacetyl hexoside (compound **34**), Kaempferol 3-*O*-rutinoside (compound **35**), and Kaempferol 3-*O*-deoxyhexosylhexoside (compound **36**) have been already characterized as components of *F. glaucescens* [25], *Andean blueberry* [28], *Rhus coriaria (Sumac)* [32], *Lonicera japonica* [34], *Potato leaves* [35], *Rapeseed petals* [36], *Echinops lanceolatus* [37], *Camellia kucha* [38]. Thus, the flavonol Kaempferol was found in extracts of *R. rugosa*, *R. davurica,* and *R. acicularis.* The CID-spectrum in positive ion modes of luteolin from extracts of *D. palmatum* is shown in Figure 7.

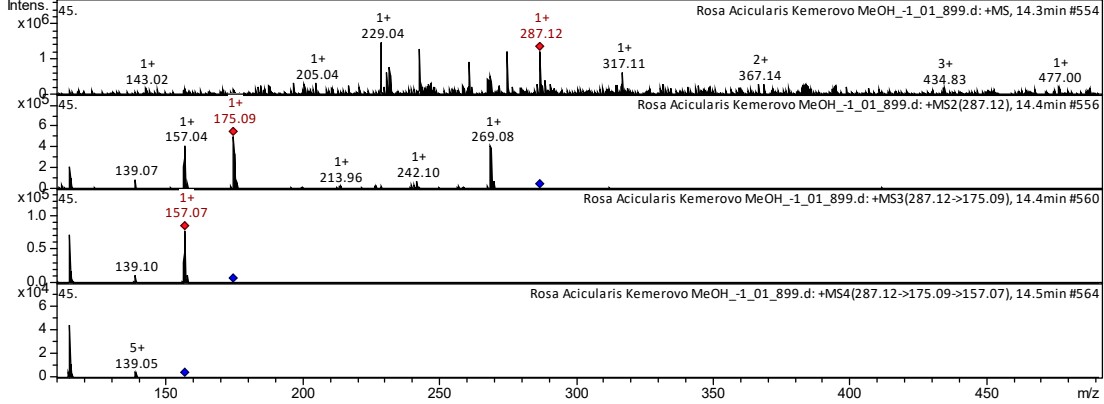

**Figure 7.** CID-spectrum of Kaempferol from extracts of *R. acicularis*, *m/z* 287.

The [M + H]⁺ ion produced six fragment ions at *m/z* 269, *m/z* 242, *m/z* 213, *m/z* 175, *m/z* 157, and *m/z* 139 (Figure 7). The fragment ion with *m/z* 175 yielded two daughter ions at *m/z* 157 and *m/z* 139. It was identified in the bibliography in extracts of *Andean blueberry* [28], *Rhus coriaria (Sumac)* [32], *Lonicera japonica* [34], and *Potato leaves* [35].

### 3.5. Pentahydroxyflavones

The flavonols Quercetin (compound **23**), Morin (compound **24**), Rhamnetin I (compound **25**), Rhamnetin II (compound **26**), Isorhamnetin (compound **27**), Avicularin (compound **31**), Taxifolin-*O*-pentoside (compound **32**), Taxifolin-3-*O*-hexoside (compound **33**), and Isorhamnetin triacetyl hexoside (compound **37**) have been already characterized as components of *Bougainvillea* [21], *Rosa rugosa* [30], *Propolis* [31], *Rhus coriaria* [32], and *Potato leaves* [35]. Thus, the flavonol Taxifolin-*O*-pentoside was found in extracts of *R. davurica*. The CID-spectrum in negative ion modes of Taxifolin-*O*-pentoside from extracts of *R. davurica* is shown in Figure 8.

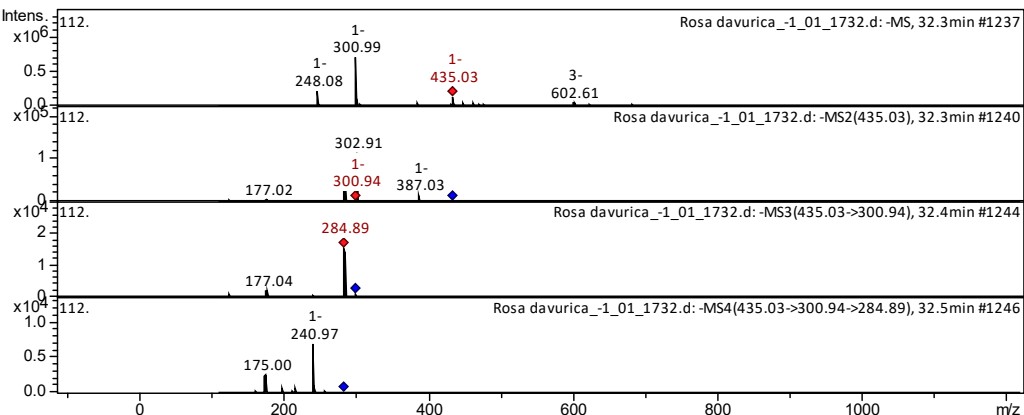

**Figure 8.** CID-spectrum of Taxifolin-*O*-pentoside from extracts of *R. davurica*, *m/z* 285.03.

The [M − H]⁻ ion produced three fragment ions at *m/z* 387, *m/z* 300, and *m/z* 177 (Figure 8). The fragment ion with *m/z* 300 yielded two daughter ions at *m/z* 284 and *m/z* 177. The fragment ion with *m/z* 284 yielded two daughter ions at *m/z* 240 and *m/z* 175. It was identified in the bibliography in extracts of *millet grains* [23] and *A. cordifolia* [25].

### 3.6. Flavan-3-ols

The flavan-3-ols Epiafzelechin (compound **38**), Catechin (compound **39**), (*epi*)Catechin (compound **40**), and Gallocatechin (compound **41**) have been characterized as components of *millet grains* [23], *G. linguiforme* [25], *Camellia kucha* [38], *strawberry, cherimoya* [39], *Rosa rugosa* [40], *Myrtle* [41], *Radix polygoni multiflori* [42], *Licania ridigna* [43], and *Rhodiola rosea* [44]. The flavan-3-ol Gallocatechin (compound **41**) was found in extract of *R. rugosa* and *R. davurica.* The CID-spectrum in negative ion modes of Gallocatechin from *R. rugosa* is shown in Figure 9.

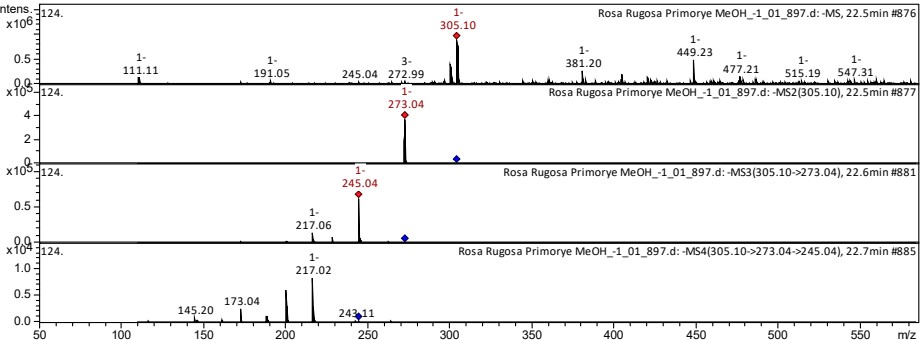

**Figure 9.** CID-spectrum of Gallocatechin from *R. rugosa*, *m/z* 305.10.

The [M − H]⁻ ion produced one fragment ion at *m/z* 273 (Figure 9). The fragment ion with *m/z* 273 yielded two daughter ions at *m/z* 269 and *m/z* 217. The fragment ion with *m/z* 245 yielded four daughter ions at *m/z* 243, *m/z* 217, *m/z* 173, and *m/z* 145. It was identified in the bibliography in extracts from *G. linguiforme* [25], *Licania ridigna* [43], and *Rhodiola rosea* [44].

### 3.7. Condensed Tannin

Prodelphinidin A-type (compound **83**) and (S)-Flavogallonic acid (compound **84**) have been already characterized as components of *Vitis vinifera* [45], *Terminalia arjuna* [46], and *R. rugosa* [47]. CID-spectrum in positive ion modes of (S)-Flavogallonic acid from *R. davurica* is shown in Figure 10. The [M + H]⁺ ion produced four fragment ions at *m/z* 453, *m/z* 407, *m/z* 321, *m/z* 247, and *m/z* 205 (Figure 10). The fragment ion with *m/z* 407 yielded three daughter ions at *m/z* 389, *m/z* 307, and *m/z* 205. This compound was identified in the bibliography in extracts from *Terminalia arjuna* [46] and *R. rugosa* [47].

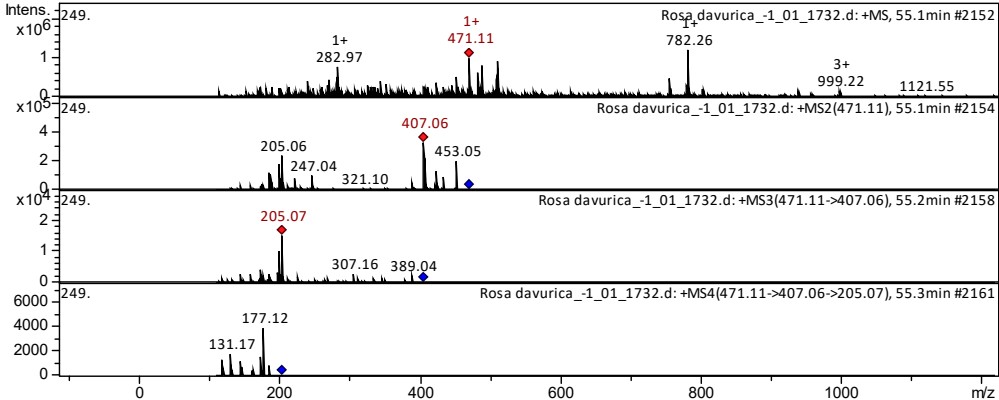

**Figure 10.** CID-spectrum of (S)-Flavogallonic acid from extracts of *R. davurica*, *m/z* 471.11.

The polyphenol composition distribution table of varieties *Rosa rugosa* Thumb., *Rosa davurica* Pall., and *Rosa acicularis* Lindl. is shown below [Table 1]. The comparison table shows the presence of some polyphenols in three types of the genus *Rosa* (kaempferol, ellagic acid). Some polyphenols are present in only one variety of the genus *Rosa*.

**Table 1.** The flavonoid composition distribution of varieties *R. rugosa* Thumb., *R. davurica* Pall., and *R. acicularis* Lindl. Blue square—presence in extracts of *R. rugosa*; red square—in extracts of *R. davurica*; green square—in extracts of *R. acicularis*.

| No. | Class of Compounds | Identified Compounds | *R. rugosa* | *R. davurica* | *R. acicularis* |
|---|---|---|---|---|---|
| 1 | Flavone | Hydroxy-methoxy (iso) flavone * | ■ | | |
| 2 | Flavone | Apigenin | | ■ | ■ |
| 3 | Flavone | Chrysoeriol [Chryseriol] * | ■ | | |
| 4 | Flavone | Hispidulin * | ■ | | |
| 5 | Flavone | 5,7-Dimethoxyluteolin * | ■ | ■ | |
| 6 | Flavone | Cirsimaritin * | | | ■ |
| 7 | Flavone | Chrysoeriol methyl ether * | | ■ | |
| 8 | Flavone | Cirsiliol * | ■ | | |
| 9 | Flavone | Tricin * | | ■ | |
| 10 | Flavone | Jaceosidin * | | ■ | |
| 11 | Flavone | 5,6,4′-Trihydroxy-7,8-dimetoxyflavone * | ■ | | |
| 12 | Flavone | Nevadensin * | | | ■ |
| 13 | Flavone | Syringetin * | | ■ | |
| 14 | Flavone | Dihydroxy-tetramethoxy(iso)flavone * | ■ | | |
| 15 | Flavone | Pentahydroxy trimethoxy flavone * | ■ | | |
| 16 | Flavone | Isovitexin * | ■ | | |
| 17 | Flavone | Genistein *C*-glucoside malonylated * | | ■ | |
| 18 | Flavone | Chrysin 6-*C*-glucoside-6″-*O*-deoxyhexoside * | ■ | | |
| 19 | Flavone | Diosmin * | | ■ | |
| 20 | Flavonol | Kaempferol | ■ | ■ | ■ |
| 21 | Flavonol | Dihydrokaempferol * | | ■ | |
| 22 | Flavonol | Isokaempferide [3-*O*-Methylkaempferol] | | ■ | |
| 23 | Flavonol | Quercetin | ■ | ■ | |
| 24 | Flavonol | Morin | | | ■ |
| 25 | Flavonol | Rhamnetin I | | | ■ |
| 26 | Flavonol | Rhamnetin II * | | | ■ |
| 27 | Flavonol | Isorhamnetin | | ■ | |
| 28 | Flavonol | Myricetin | | ■ | |
| 29 | Flavonol | Mearnsetin * | | ■ | |
| 30 | Flavonol | Kaempferol-3-*O*-α-L-rhamnoside * | ■ | | |
| 31 | Flavonol | Avicularin | | | ■ |
| 32 | Flavonol | Taxifolin-*O*-pentoside * | | ■ | |
| 33 | Flavonol | Taxifolin-3-*O*-hexoside * | | | ■ |
| 34 | Flavonol | Kaempferol diacetyl hexoside | | ■ | |
| 35 | Flavonol | Kaempferol 3-*O*-rutinoside | | ■ | |
| 36 | Flavonol | Kaempferol 3-*O*-deoxyhexosylhexoside | | | ■ |
| 37 | Flavonol | Isorhamnetin triacetyl hexoside * | | | ■ |
| 38 | Flavan-3-ol | Epiafzelechin [(epi)Afzelechin] * | | ■ | |
| 39 | Flavan-3-ol | Catechin [D-Catechol] | | ■ | ■ |
| 40 | Flavan-3-ol | (+)-Epicatechin * | | ■ | |

**Table 1.** *Cont.*

| No. | Class of Compounds | Identified Compounds | *R. rugosa* | *R. davurica* | *R. acicularis* |
|---|---|---|---|---|---|
| 41 | Flavan-3-ol | **Gallocatechin** [+(-)Gallocatechin] * | ▰ | ▰ | |
| 42 | Flavanone | **Naringenin** [Naringetol; Naringenine] | ▰ | | |
| 43 | Flavanone | **Fustin** [2,3-Dihydrofistein] * | | ▰ | |
| 44 | Flavanone | **Eriodictyol** [3′,4′,5,7-tetrahydroxy-flavanone] | | | ▰ |
| 45 | Flavanone | **Eriodictyol-7-*O*-glucoside** * | ▰ | | |
| 46 | Hydroxycinnamic acid | **Caffeic acid** * | ▰ | | ▰ |
| 47 | Phenolic acid | **Quinic acid** | ▰ | ▰ | |
| 48 | Phenolic acid | **Citric acid** [Anhydrous; Citrate] * | ▰ | ▰ | |
| 49 | Phenolic acid | ***trans*-Ferulic acid** | ▰ | | |
| 50 | Phenolic acid | **Hydroxy methoxy dimethylbenzoic acid** * | | ▰ | |
| 51 | Phenolic acid | **Syringic acid** | ▰ | | ▰ |
| 52 | Phenolic acid | **3,3,4,4-Tetrahydroxy-5-oxo-cyclohexanecarboxylic acid** * | ▰ | | |
| 53 | Phenolic acid | **Hydroxyferulic acid** * | | ▰ | |
| 54 | Hydroxycinnamic acid | **Sinapic acid** [trans-Sinapic acid] | | ▰ | |
| 55 | Phenolic acid | **2,4,6-Trihydroxy-3,5-dimethoxybenzoic acid** * | | | ▰ |
| 56 | Hydroxybenzoic acid | **Ellagic acid** * | ▰ | ▰ | ▰ |
| 58 | Phenolic acid | ***p*-Coumaroylquinic acid** * | | ▰ | |
| 58 | Phenolic acid | **Ginkgoic acid** * | ▰ | | |
| 59 | Phenolic acid | **1-[(Acetyl-L-cysteinyl)oxy]-2,3,4,5-tetrahydroxycyclohexane-1-carboxylic acid** * | ▰ | | |
| 60 | Phenolic acid | **Chlorogenic acid** [3-*O*-Caffeoylquinic acid] * | | | ▰ |
| 61 | Phenolic acid | **Neochlorogenic acid** [5-*O*-Caffeoylquinic acid] | ▰ | | |
| 62 | Phenolic acid | **Rosmarinic acid** | ▰ | | |
| 63 | Phenolic acid | **5-Hydroxy feruloyl hexose** * | | ▰ | |
| 64 | Phenolic acid | **Salvianolic acid D** * | ▰ | | |
| 65 | Phenolic acid | **Salvianolic acid B** [Danfensuan B] * | | ▰ | |
| 66 | Stilbene | **Pinosylvin** * | ▰ | | |
| 67 | Stilbene | **Resveratrol** * | ▰ | | ▰ |
| 68 | Stilbene | **3-Hydroxyresveratrol** * | ▰ | | ▰ |
| 69 | Lignan | **Pinoresinol** * | ▰ | | |
| 70 | Lignan | **Arctigenin** * | | | ▰ |
| 71 | Coumarin | **3,4,5-Trimethoxycoumarin** * | ▰ | | |
| 72 | Coumarin | **Fraxin** (Fraxetin-8-*O*-glucoside) * | ▰ | | |
| 73 | Anthocyanidin | **Anthocyanidin** [cyanidin chloride; Cyanidin] * | ▰ | | |
| 74 | Anthocyanidin | **Petunidin** * | ▰ | | ▰ |
| 75 | Anthocyanidin | **Cyanidin-3-*O*-glucoside** [Cyanidin 3-*O*-beta-D-Glucoside; Kuromarin] * | | ▰ | |
| 76 | Anthocyanidin | **Delphinidin *O*-pentoside** * | | ▰ | |
| 77 | Anthocyanidin | **Pelargonidin 3-*O*-(6-*O*-malonyl-beta-D-glucoside)** * | | ▰ | |
| 78 | Anthocyanidin | **Cyanidin 3-(6″-Succinyl-Glucoside)** * | ▰ | | ▰ |
| 79 | Anthocyanidin | **Delphinidin malonyl hexoside** * | | ▰ | |
| 80 | Anthocyanidin | **Cyanidin-3-*O*-dioxayl-glucoside** * | ▰ | | |
| 81 | Anthocyanidin | **Delphinidin 3,5-dihexoside** * | | ▰ | |
| 82 | Tannin | **Prodelphinidin A-type** * | | | ▰ |
| 83 | Hydrolysable tannin | **(S)-Flavogallonic acid** | | ▰ | |

**Table 1.** *Cont.*

| No. | Class of Compounds | Identified Compounds | *R. rugosa* | *R. davurica* | *R. acicularis* |
|---|---|---|---|---|---|
| 84 | Ellagitannin | **Punicalin alpha \*** | | 🟥 | |
| 85 | Phenylpropanoid | **Coniferin \*** | | 🟥 | |
| 86 | Gallate ester | **Ethyl gallate \*** | | | 🟩 |
| 87 | Gallate ester | **Beta-Glucogallin \*** | 🟦 | | 🟩 |
| 88 | Dihydrochalcone | **Phloretin** [Dihydronaringenin; Phloretol] \* | 🟦 | 🟥 | 🟩 |
| 89 | Flavonoid | **Diphylloside B \*** | | | 🟩 |
| 90 | Flavonoid | **Demethylanhydroicaritin-7-*O*-glucopyranosyl-3-*O*-acetylated rhamnopyranosyl-xylopyranoside \*** | | | 🟩 |

\* Polyphenols identified for the first time in genus *Rosa.*

The following polyphenols are present in only *R. rugosa:* Hydroxy-methoxy (iso)flavone, Chrysoeriol, Hispidulin, Cirsiliol, 5,6,4′-Trihydroxy-7,8-dimetoxyflavone, Dihydroxy-tetramethoxy (iso)flavone, Pentahydroxy trimethoxy flavone, Isovitexin, Chrysin 6-*C*-glucoside-6″-*O*-deoxyhexoside, Kaempferol-3-*O*-α-L-rhamnoside, Naringenin, Eriodictyol-7-*O*-glucoside, *trans*-Ferulic acid, 3,3,4,4-Tetrahydroxy-5-oxo-cyclohexanecarboxylic acid, Ginkgoic acid, 1-[(Acetyl-L-cysteinyl)oxy]-2,3,4,5-tetrahydroxycyclohexane-1-carboxylic acid, Neochlorogenic acid, Rosmarinic acid, Salvianolic acid D, Pinosylvin, Pinoresinol, 3,4,5-rimethoxy coumarin, Fraxin, Anthocyanidin [cyanidin chloride; Cyanidin], Cyanidin-3-*O*-dioxayl-glucoside.

The following polyphenols are present in only *R. davurica*—Chrysoeriol methyl ether, Tricin, Jaceosidin, Syringetin, Genistein C-glucoside malonylated, Diosmin, Dihydrokaempferol, Isokaempferide, Isorhamnetin, Myricetin, Mearnsetin, Taxifolin-*O*-pentoside, Kaempferol diacetyl hexoside, Kaempferol 3-*O*-rutinoside, Epiafzelechin, (*epi*)Catechin, Fustin, Hydroxy methoxy dimethylbenzoic acid, Hydroxyferulic acid, Sinapic acid, *p*-Coumaroylquinic acid, Salvianolic acid B, Cyanidin-3-*O*-glucoside, Delphinidin-*O*-pentoside, Pelargonidin-3-*O*-(6-*O*-malonyl-beta-D-glucoside), Delphinidin malonyl hexoside, Delphinidin 3,5-dihexoside, (S)-Flavogallonic acid, Punicalin alpha, Coniferin.

The following polyphenols are present in only *R. acicularis*—Cirsimaritin, Nevadensin, Morin, Rhamnetin I, Rhamnetin II, Nevadensin, Taxifolin-3-*O*-hexoside, Kaempferol 3-*O*-deoxyhexosylhexoside, Isorhamnetin triacetyl hexoside, Eriodictyol, 2,4,6-Trihydroxy-3,5-dimethoxybenzoic acid, Arctigenin, Prodelphinidin A-type, Ethyl gallate, Diphylloside B.

Thus, 146 metabolome compounds were identified in the extracts of *R. rugosa, R. davurica*, and *R. acicularis*, many of which are characteristic of the genus *Rosa* (family Rosaceae). Of these, 115 components were identified for the first time in the genus *Rosa*. These are flavones: Chrysoeriol, Hispidulin, 5,7-Dimethoxyluteolin, Cirsimaritin, Cirsiliol, Tricin, Jaceosidin, Nevadensin, Syringetin, Isovitexin, Genistein C-glucoside malonylated, Chrysin 6-*C*-glucoside-6″-*O*-deoxyhexoside; flavanols: Dihydrokaempferol, Rhamnetin II, Kaempferol-3-*O*-α-L-rhamnoside, Taxifolin-*O*-pentoside, Taxifolin-3-*O*-hexoside, Isorhamnetin triacetyl hexoside; flavan-3-ols: Epiafzelechin and Gallocatechin; flavanones: Naringenin, Fustin; phenolic acids: Caffeic acid, Citric acid, Hydroxy methoxy dimethylbenzoic acid, Hydroxyferulic acid, Ellagic acid, *p*-Coumaroylquinic acid, Ginkgoic acid, Salvianolic acid D, Salvianolic acid B; stilbenes: Pinosylvin, Resveratrol, 3-Hydroxyresveratrol; lignans: Pinoresinol, Arctigenin; coumarins: 3,4,5–Trimethoxycoumarin, Fraxin; anthocyanins Cyanidin 3-*O*-glucoside, Delphinidin *O*-pentoside, Pelargonidin 3-*O*-(6-*O*-malonyl-β-D-glucoside), Cyanidin 3-(6″-Succinyl-Glucoside), Delphinidin malonyl hexoside, Cyanidin 3-*O*-dioxayl-glucoside, Delphinidin 3,5-dihexoside, etc.

## 4. Materials and Methods

### 4.1. Materials

Aboveground phyto *Rosa rugosa* Thumb., *Rosa davurica* Pall., and *Rosa acicularis* Lindl. collected during expedition work on the territory of the Russian Far East, Trans-Baikal Region, and Western Siberia during the period of ripening (July–September, 2020). Phyto mass of *R. davurica* was collected on the territory of Buryatia (N 52°21′97″ E 108°59′84″), in September 2020. Phyto mass of *R. rugosa* was collected on the territory of Primorsky Krai, Russia (N 42°36′10″ E 131°10′55″), during the period from 10 to 20 August, 2020. Phyto mass of *R. acicularis* was collected on the territory of Kemerovo, Western Siberia (N 55°21′15″ E 86°05′23″), in August 2020. All samples were morphologically authenticated according to the current standard of Pharmacopoeia of the Eurasian Economic Union [48].

The results were obtained using the equipment of the Center for Collective Use of Scientific Equipment of TSU named after G.R. Derzhavin.

### 4.2. Chemicals and Reagents

HPLC-grade acetonitrile was purchased from Fisher Scientific (Southborough, UK), MS-grade formic acid was from Sigma-Aldrich (Steinheim, Germany). Ultra-pure water was prepared from a SIEMENS ULTRA clear (SIEMENS water technologies, Germany), and all other chemicals were analytical grade.

### 4.3. Fractional Maceration

To obtain highly concentrated extracts, fractional maceration was applied. In this case, the total amount of the extractant (methyl alcohol of reagent grade) is divided into 3 parts and is consistently infused on potato with the first part, then with the second and third. The infusion time of each part of the extractant was 7 days.

### 4.4. Liquid Chromatography

HPLC was performed using Shimadzu LC-20 Prominence HPLC (Shimadzu, Japan), equipped with an UV-sensor and a Shodex ODP-40 4E reverse phase column to perform the separation of multicomponent mixtures. The gradient elution program was as follows: 0.01–5 min, 100% $CH_3CN$; 5–45 min, 100–25% $CH_3CN$; 45–55 min, 25–0% $CH_3CN$; control washing: 55–60 min, 0% $CH_3CN$. The entire HPLC analysis was conducted with an ESI detector at wavelengths of 230 ηm and 330 ηm; the temperature corresponded to 17 °C. The injection volume was 1 mL.

### 4.5. Mass Spectrometry

MS analysis was performed on an ion trap amaZon SL (BRUKER DALTONIKS, Germany) equipped with an ESI source in negative and positive ion modes. The optimized parameters were obtained as follows: ionization source temperature: 70 °C, gas flow: 4 L/min, nebulizer gas (atomizer): 7.3 psi, capillary voltage: 4500 V, end plate bend voltage: 1500 V, fragmentary: 280 V, collision energy: 60 eV. A four-stage ion separation mode (MS/MS mode) was implemented.

## 5. Conclusions

The extracts of *Rosa rugosa* Thumb., *Rosa davurica* Pall., and *Rosa acicularis* Lindl. contain a large number of polyphenolic complexes which are biologically active compounds. For the most complete and safe extraction, the method of maceration with MeOH was used. To identify target analytes in extracts, HPLC was used in combination with the ion trap. The results of a preliminary study showed the presence of 146 bioactive compounds, of which 115 were identified for the first time in the genus *Rosa* (family *Rosaceae*). Of these 115 chemical compounds identified for the first time in the genus *Rosa*, 70 compounds belonged to the polyphenolic group: 18 flavones, 7 flavonols, 3 flavan-3-ols, 2 flavanones, 14 phenolic acids, 3 stilbenes, 2 lignans, 2 coumarins, 9 anthocyanidins, 3 tannins, etc. The proven richness of the bioactive components of targeted extracts of *R. rugosa*, *R. davurica*,

and *R. acicularis* invites extensive biotechnological and pharmaceutical research, which can make a significant contribution both in the field of functional and enriched nutrition, and in the field of cosmetology and pharmacy. It should also be noted that the variability of the genus *Rosa* (family *Rosaceae*) contributes to the selection of the most drought-resistant species and samples for household, decorative, and forest reclamation needs in the arid climatic zones of Eurasia.

It is important to note that the useful properties of the genus *Rosa* (family *Rosaceae*) are: food (*R. rugosa, R. acicularis*), perfumery (*R. acicularis, R. ecae*), nectariferous (*R. canina, R. cinnamomea*), decorative (*R. acicularis, R. rugosa*), and soil-strengthening (*R. acicularis, R. rugosa, R. spinosissima*). A wide variety of biologically active polyphenolic compounds opens up rich opportunities for the creation of new drugs, as well as bioactive additives based on extracts from the genus *Rosa*.

**Author Contributions:** Conceptualization, B.A.B., A.N.K. and M.P.R.; methodology, Y.Y.Z., A.G.B. and M.P.R.; software, M.P.R.; validation, A.N.K., M.P.R. and K.S.G.; formal analysis, M.P.R. and A.M.Z.; investigation, A.S.S. and S.E.; resources, K.S.G., B.A.B., and Y.Y.Z.; data curation, B.A.B.; writing—original draft preparation—M.P.R. and A.M.Z.; writing—review and editing A.M.Z. and K.S.G.; visualization, M.P.R. and A.M.Z.; supervision, K.S.G.; project administration, B.A.B., K.S.G. and S.E. All authors have read and agreed to the published version of the manuscript.

**Funding:** The work was carried out with the support of the grant Young Scientists ESSTUM 2022 and according to No. 0662-2019-0003, "Genetic resources of vegetable and melons of the world collection of N.I. Vavilov All-Russian Institute of Plant Genetic Resources: effective ways of expanding diversity, disclosing the patterns of hereditary variability, use of adaptive potential".

**Institutional Review Board Statement:** No applicable.

**Informed Consent Statement:** No applicable.

**Data Availability Statement:** No applicable.

**Acknowledgments:** Research work according to No. 0662-2019-0003 "Genetic resources of vegetable and melons of the world collection of N.I. Vavilov All-Russian Institute of Plant Genetic Resources: effective ways of expanding diversity, disclosing the patterns of hereditary variability, use of adaptive potential".

**Conflicts of Interest:** The authors declare no conflict of interest.

# Appendix A

**Table A1.** Compounds identified from the extracts of *Rosa rugosa* Thumb., *Rosa davurica* Pall., and *Rosa acicularis* Lindl. in positive and negative ionization modes by HPLC–ion trap–MS/MS.

| No. | Class of Compounds | Identified Compounds | Formula | Mass | Molecular Ion [M − H]⁻ | Molecular Ion [M + H]⁺ | 2 Fragmentation MS/MS | 3 Fragmentation MS/MS | 4 Fragmentation MS/MS | References |
|---|---|---|---|---|---|---|---|---|---|---|
| | | **POLYPHENOLS** | | | | | | | | |
| 1 | Flavone | **Hydroxy-methoxy (iso) flavone** * | $C_{16}H_{12}O_4$ | 268.2641 | | 269 | 252 | 221 | 190 | Propolis [31] |
| 2 | Flavone; | **Apigenin** [5,7-Dixydroxy-2-(40Hydroxyphenyl)-4H-Chromen-4-One] | $C_{15}H_{10}O_5$ | 270.2369 | | 271 | 253 | 224 | | *Hedyotis diffusa* [27]; Andean blueberry [28]; *Stevia rebaudiana* [29]; *Rosa rugosa* [30]; Propolis [31] |
| 3 | Flavone | **Chrysoeriol** [Chryseriol] * | $C_{16}H_{12}O_6$ | 300.2629 | | 301 | 269; 195 | 241 | | Mentha [25]; Propolis [31]; *Rhus coriaria* [32]; Mexican lupine species [33] |
| 4 | Flavone | **Hispidulin** * | $C_{16}H_{12}O_6$ | 300.2629 | | 301 | 269; 241; 197 | 224; 180; 153 | | Mentha [25]; *Cirsium japonicum* [49] |
| 5 | Flavone | **5,7-Dimethoxyluteolin** * | $C_{17}H_{14}O_6$ | 314.2895 | 313 | | 212; 285; 184; 113 | 113; 145; 185 | | *Syzygium aromaticum* [18] |
| 6 | Flavone | **Cirsimaritin** [Scrophulein; 4′,5-Dihydroxy-6,7-Dimethoxyflavone; 7-Methylcapillarisin] * | $C_{17}H_{14}O_6$ | 314.2895 | | 315 | 300; 240; 213; 185 | 272; 227; 185; 168; 135 | 185 | *Ocimum* [19]; *Rosmarinus officinalis* [20] |
| 7 | Flavone | **Chrysoeriol methyl ether** * | $C_{17}H_{14}O_6$ | 314.2895 | | 315 | 287; 241; 187 | 187 | 169 | Bougainvillea [21] |
| 8 | Flavone | **Cirsiliol** * | $C_{17}H_{14}O_7$ | 330.2889 | 329 | | 229 | 211; 127 | 209; 125 | *Ocimum* [19] |
| 9 | Flavone | **Tricin** [5,7,4′-trihydroxy-3′,5′-dimetoxyflavone] * | $C_{17}H_{14}O_7$ | 330.2889 | 329 | | 314; 259; 229 | 299; 271 | 271; 227 | *Triticum aestivum* [22]; millet grains [23]; *Sasa veitchii; Phyllostachys nigra* [24] |
| 10 | Flavone | **Jaceosidin** [5,7,4′-trihydroxy-6′,5′-dimetoxyflavone] * | $C_{17}H_{14}O_7$ | 330.2889 | | 331 | 303; 185 | | | Mentha [26,50] |
| 11 | Flavone | **5,6,4′-Trihydroxy-7,8-dimetoxyflavone** * | $C_{17}H_{14}O_7$ | 330.2889 | | 331 | 299; 179 | 211 | | *F. glaucescens; F. herrerae* [25]; Mentha [26] |
| 12 | Flavone | **Nevadensin** * | $C_{18}H_{16}O_7$ | 344.3154 | | 345 | 330 | 315 | 286; 259; 183; 133 | *Ocimum* [19]; *Mentha* [26] |
| 13 | Flavone | **Syringetin** * | $C_{17}H_{14}O_8$ | 346.2883 | | 347 | 317; 218 | 289; 218 | 261; 191 | *C. edulis* [25] |
| 14 | Flavone | **Dihydroxy-tetramethoxy(iso)flavone** * | $C_{19}H_{18}O_8$ | 374.3414 | | 375 | 343 | 315 | 225 | Propolis [31] |
| 15 | Flavone | **Pentahydroxy trimethoxy flavone** * | $C_{18}H_{16}O_{10}$ | 392.3136 | | 393 | 377; 375; 275; 213 | 357 | 329; 286 | *F. glaucescens; C. edulis* [25] |
| 16 | Flavone | **Isovitexin** [Saponaretin; Homovitexin; Apigenin-6-C-Glucoside] * | $C_{21}H_{20}O_{10}$ | 432.3775 | | 433 | 415; 335; 243; 175 | 261; 243; 191; 155 | 135 | millet grains [23]; *Phyllostachys nigra* [24]; *Rhus coriaria* [32] |
| 17 | Flavone | **Genistein C-glucoside malonylated** * | $C_{24}H_{22}O_{13}$ | 518.4237 | 517 | | 473; 455 | 455; 413; 339 | 425 | Mexican lupine species [33] |
| 18 | Flavone | **Chrysin 6-C-glucoside-6″-O-deoxyhexoside** * | $C_{27}H_{30}O_{13}$ | 562.5193 | | 563 | 400; 363; 305; 239 | 130; 162; 191; 214 | | *Passiflora incarnata* [51] |
| 19 | Flavone | **Diosmin** [Diosmetin-7-O-rutinoside; Barosmin; Diosimin] * | $C_{28}H_{32}O_{15}$ | 608.5447 | | 609 | 591; 429; 355; 269 | 285 | 269 | *F. glaucescens* [25]; *Mentha* [26]; Lemon [39]; *Grataegi Fructus* [52] |
| 20 | Flavonol | **Kaempferol** [3,5,7-Trihydroxy-2-(4-hydroxyphenyl)-4H-chromen-4-one] | $C_{15}H_{10}O_6$ | 286.2363 | | 287 | 187; 227 | 189; 125 | | Andean blueberry [28]; *Rhus coriaria* (Sumac) [32]; *Lonicera japonica* [34]; Potato leaves [35]; Rapeseed petals [36] |
| 21 | Flavonol | **Dihydrokaempferol** [Aromadendrin; Katuranin] * | $C_{15}H_{12}O_6$ | 288.2522 | 287 | | 259; 185; 117 | 215 | 197 | *F. glaucescens* [25]; Andean blueberry [28]; *Echinops lanceolatus* [37]; *Camellia kucha* [38] |

**Table A1.** *Cont.*

| No. | Class of Compounds | Identified Compounds | Formula | Mass | Molecular Ion [M − H]⁻ | Molecular Ion [M + H]⁺ | 2 Fragmentation MS/MS | 3 Fragmentation MS/MS | 4 Fragmentation MS/MS | References |
|---|---|---|---|---|---|---|---|---|---|---|
| 22 | Flavonol | **Isokaempferide** [3-O-Methylkaempferol] | $C_{16}H_{12}O_6$ | 300.2629 | | 301 | 300; 274; 257; 212; 184 | 286; 242; 201 | 240 | *Rosa rugosa* [30]; Propolis [31] |
| 23 | Flavonol | **Quercetin** | $C_{15}H_{10}O_7$ | 302.2357 | | 303 | 285 | 257 | 201;117 | Bougainvillea [21]; Propolis [31]; *Rosa rugosa* [30]; *Rhus coriaria* [32]; Potato leaves [35] |
| 24 | Flavonol | **Morin** [Aurantica; Calico Yellow; Toxylon Pomiferum; 2′,3,4′,5,7-Pentahydroxyflavone] | $C_{15}H_{10}O_7$ | 302.2357 | 301 | | 283; 265; 221 | 221 | 203; 151; 127 | *Rosa rugosa* [30]; Red wines [53] |
| 25 | Flavonol | **Rhamnetin I** [beta-Rhamnocitrin; Quercetin 7-Methyl ether] | $C_{16}H_{12}O_7$ | 316.2623 | | 317 | 299; 269; 233; 185; 165 | 147; 123 | | *Rosa rugosa* [30]; *Rhus coriaria* L. (Sumac) [32] |
| 26 | Flavonol | **Rhamnetin II** * | $C_{16}H_{12}O_7$ | 316.2623 | | 317 | 165;185; 155; 123 | 147; 123 | 119 | *Syzygium aromaticum* [18]; Propolis [31]; *Rhus coriaria* L. (Sumac) [32]; *Spondias purpurea* [54] |
| 27 | Flavonol | **Isorhamnetin** [Isorhamnetol; Quercetin 3′-Methyl ether; 3-Methylquercetin] | $C_{16}H_{12}O_7$ | 316.2623 | | 317 | 285; 234; 190; 156 | 256; 214 | 229; 201 | *Rosmarinus officinalis* [20]; Andean blueberry [28]; *Rosa rugosa* [30]; Propolis [31]; *Vaccinium macrocarpon* [55]; *Embelia* [56] |
| 28 | Flavonol | **Myricetin** | $C_{15}H_{10}O_8$ | 318.2351 | | 319 | 289; 217; 185 | 261; 191 | | millet grains [23]; *F. glaucescens* [25]; Andean blueberry [28]; *Rosa rugosa* [30]; Propolis [31]; *Vaccinium macrocarpon* [55] |
| 29 | Flavonol | **Mearnsetin** * | $C_{16}H_{12}O_8$ | 332.2617 | 331 | | 287 | 259 | 215; 187; 159 | *Eucalyptus* [57] |
| 30 | Flavonol | **Kaempferol-3-O-α-ʟ-rhamnoside** * | $C_{21}H_{20}O_{10}$ | 432.3775 | | 433 | 415; 313; 241; 195 | 123; 257; 239 | | *C.edulis; F. glaucescens* [25]; *Rhus coriaria* [32]; *Cassia abbreviata* [58]; *Euphorbia hirta* [59] |
| 31 | Flavonol | **Avicularin** (Quercetin 3-Alpha-ʟ-Arabinofuranoside; Avicularoside) | $C_{20}H_{18}O_{11}$ | 434.3503 | 433 | | 301 | 273; 229; 192; 179; 151 | 169; 151 | Propolis [31]; *Eucalyptus Globulus* [60]; *Rosa rugosa* [61] |
| 32 | Flavonol | **Taxifolin-O-pentoside** [Dihydroquercetin pentoside] * | $C_{20}H_{20}O_{11}$ | 436.371 | 435 | 301; 177 | 285; 177 | 241; 175 | | millet grains [23]; *A. cordifolia* [25] |
| 33 | Flavonol | **Taxifolin-3-O-hexoside** [Dihydroquercetin-3-O-hexoside] * | $C_{21}H_{22}O_{12}$ | 466.3922 | | 467 | 287; 305; 334; 449 | 268; 256; 227; 202 | | millet grains [23]; Andean blueberry [28]; *Euphorbia hirta* [59]; *Rubus ulmifolius* [62] |
| 34 | Flavonol | **Kaempferol diacetyl hexoside** | $C_{25}H_{24}O_{13}$ | 532.4503 | | 533 | 432; 531; 289 | 415; 295 | 385 | *A. cordifolia* [25] |
| 35 | Flavonol | **Kaempferol 3-O-rutinoside** | $C_{27}H_{30}O_{15}$ | 594.5181 | | 595 | 285; 165 | 165 | | *Rhus coriaria* [32]; *Lonicera japonica* [34]; *Camellia kucha* [38]; Strawberry [39] |
| 36 | Flavonol | **Kaempferol 3-O-deoxyhexosylhexoside** | $C_{27}H_{30}O_{15}$ | 594.5181 | | 595 | 287; 263; 165 | 213; 197; 165 | 157; 145 | *Stevia rebaudiana* [29]; *Rosa rugosa* [40]; *Spondias purpurea* [54] |
| 37 | Flavonol | **Isorhamnetin triacetyl hexoside** * | $C_{28}H_{28}O_{15}$ | 604.5129 | | 605 | 443; 417; 317; 279 | 329; 311; 255; 211 | | *A. cordifolia* [25] |
| 38 | Flavan-3-ol | **Epiafzelechin** [(epi)Afzelechin] * | $C_{15}H_{14}O_5$ | 274.2687 | | 275 | 244; 157 | 157; 215 | 127 | *A. cordifolia; F. glaucescens; F. herrerae* [25]; *Cassia abbreviata* [58]; *Cassia granidis* [63] |
| 39 | Flavan-3-ol | **Catechin** [D-Catechol] | $C_{15}H_{14}O_6$ | 290.2681 | | 291 | 272; 174 | 245 | 198 | millet grains [23]; *C. edulis* [25]; *Camellia kucha* [38]; strawberry, cherimoya [39]; *Rosa rugosa* [40]; Myrtle [41]; *Radix polygoni multiflori* [42]; *Rosa rugosa* [64] |
| 40 | Flavan-3-ol | **(epi)Catechin** * | $C_{15}H_{14}O_6$ | 290.2681 | | 291 | 273; 117 | 255; 145 | | Andean blueberry [28]; *C. edulis* [25]; *Camellia kucha* [38]; *Radix polygoni multiflori* [42] |
| 41 | Flavan-3-ol | **Gallocatechin** [+(-)Gallocatechin] * | $C_{15}H_{14}O_7$ | 306.2675 | | 307 | 291 | 263; 189 | 206 | *G. linguiforme* [25]; *Licania ridigna* [43]; *Rhodiola rosea* [44] |

**Table A1.** *Cont.*

| No. | Class of Compounds | Identified Compounds | Formula | Mass | Molecular Ion [M − H]⁻ | Molecular Ion [M + H]⁺ | 2 Fragmentation MS/MS | 3 Fragmentation MS/MS | 4 Fragmentation MS/MS | References |
|---|---|---|---|---|---|---|---|---|---|---|
| 42 | Flavanone | **Naringenin** [Naringetol; Naringenine] | $C_{15}H_{12}O_5$ | 272.5228 | | 273 | 153; 256 | 125 | | *G. linguiforme* [25]; Andean blueberry [28]; *Stevia rebaudiana* [29]; *Rosa rugosa* [30]; *Mexican lupine species* [33]; Rapeseed petals [36]; *Punica granatum* [65] |
| 43 | Flavanone | **Fustin** [2,3-Dihydrofistein] * | $C_{15}H_{12}O_6$ | 288.2522 | | 289 | 269; 140 | 179 | | *F. glaucescens*; *F. pottsii* [25] |
| 44 | Flavanone | **Eriodictyol** [3′,4′,5,7-tetrahydroxy-flavanone] | $C_{15}H_{12}O_6$ | 288.2522 | 287 | | 269; 241; 155; 127 | 267; 251; 223; 183; 155 | 249; 199; 155 | *Rosmarinus officinalis* [20]; Andean blueberry [28]; *Rosa rugosa* [30]; Propolis [31]; *Embelia* [56] |
| 45 | Flavanone | **Eriodictyol-7-O-glucoside** [Pyracanthoside; Miscanthoside] * | $C_{21}H_{22}O_{11}$ | 450.3928 | 449 | | 269; 151 | 225 | | *Impatiens glandulifera* Royle [66] |
| 46 | Hydroxycinnamic acid | **Caffeic acid** * | $C_9H_8O_4$ | 180.1574 | | 181 | 135 | 119 | | *Triticum* [22]; millet grains [23]; *Lonicera japonica* [24]; *Radix polygoni multiflori* [42]; Mentha [50]; *Malva sylvestris* [67] |
| 47 | Phenolic acid | **Quinic acid** | $C_7H_{12}O_6$ | 192.1666 | | 193 | 191; 147 | 173; 136 | 131 | Andean blueberry [28]; *Stevia rebaudiana* [29]; *Rhus coriaria* [32]; *Lonicera japonicum* [34]; *Camellia kucha* [38]; *Rosa rugosa* [40] |
| 48 | Phenolic acid | **Citric acid** [Anhydrous; Citrate] * | $C_6H_8O_7$ | 192.1235 | 191 | | 111; 173 | 111 | | *Stevia rebaudiana* [29]; Potato leaves [35]; Strawberry, Lemon, Cherimoya, Papaya, Passion fruit [39]; Mentha [50]; *Punica granatum* [65] |
| 49 | Phenolic acid | *trans*-**Ferulic acid** | $C_{10}H_{10}O_4$ | 194.184 | | 195 | 153 | 125 | | millet grains [23]; *Rosa rugosa* [30]; *Sanguisorba officinalis* [68] |
| 50 | Phenolic acid | **Hydroxy methoxy dimethylbenzoic acid** * | $C_{10}H_{12}O_4$ | 196.1999 | 195 | | 129; 177 | | | *F. herrerae*; *F. glaucescens* [25] |
| 51 | Phenolic acid | **Syringic acid** | $C_9H_{10}O_5$ | 198.1727 | | 199 | 157; 183; 119 | 142 | | *Bougainvillea* [21]; millet grains [29]; *A. cordifolia*; *G. linguiforme*; *F. glaucescens* [25]; *Rosa rugosa* [30]; *Actinidia* [69] |
| 52 | Phenolic acid | **3,3,4,4-Tetrahydroxy-5-oxo-cyclohexanecarboxylic acid** * | $C_7H_{10}O_7$ | 206.1501 | | 207 | 161; 189 | 143 | 119 | *Actinidia* [69] |
| 53 | Phenolic acid | **Hydroxyferulic acid** * | $C_{10}H_{10}O_5$ | 210.1834 | | 211 | 193 | 175 | 133 | Andean blueberry [28] |
| 54 | Hydroxycinnamic acid | **Sinapic acid** [trans-Sinapic acid] | $C_{11}H_{12}O_5$ | 224.2100 | | 225 | 209 | 139; 192 | | millet grains [23]; Andean blueberry [28]; *Rosa rugosa* [30]; Rapeseed petals [36]; *Cherimoya* [39] |
| 55 | Phenolic acid | **2,4,6-Trihydroxy-3,5-dimethoxybenzoic acid** * | $C_9H_{10}O_7$ | 230.1715 | | 231 | 229; 211; 185; 155 | 168; 143 | 127 | *Actinidia* [69] |
| 56 | Hydroxybenzoic acid | **Ellagic acid** [Benzoaric acid; Elagostasine; Lagistase; Eleagic acid] * | $C_{14}H_6O_8$ | 302.1926 | 301 | | 256 | 185 | | *Rhus coriaria* [32]; *Eucalyptus* [57]; *Eucalyptus Globulus* [60] |
| 57 | Phenolic acid | *p*-**Coumaroylquinic acid** * | $C_{16}H_{18}O_8$ | 338.3093 | | 339 | 191; 320; 252 | 149 | | Andean blueberry [28]; *F. glaucescens* [25]; *Eucalyptus Globulus* [60]; *Actinidia* [69] |
| 58 | Phenolic acid | **Ginkgoic acid** [Ginkgolic acid; Romanicardic acid] * | $C_{22}H_{34}O_3$ | 346.5036 | | 347 | 301; 130 | 130 | | Propolis [31] |
| 59 | Phenolic acid | **1-[(Acetyl-L-cysteinyl)oxy]-2,3,4,5-tetrahydroxycyclohexane-1-carboxylic acid** * | $C_{12}H_{19}O_9NS$ | 353.3456 | | 354 | 335 | 192; 286 | 132; 176 | *Actinidia* [69] |
| 60 | Phenolic acid | **Chlorogenic acid** [3-O-Caffeoylquinic acid] * | $C_{16}H_{18}O_9$ | 354.3087 | 353 | | 191 | 173 | | *Bougainvillea* [21]; Andean blueberry [28]; *Rhus coriaria* [32]; *Lonicera japonicum* [34]; Potato leaves [35]; Rapeseed petals [28]; *Vaccinium macrocarpon* [55] |

**Table A1.** *Cont.*

| No. | Class of Compounds | Identified Compounds | Formula | Mass | Molecular Ion [M − H]⁻ | Molecular Ion [M + H]⁺ | 2 Fragmentation MS/MS | 3 Fragmentation MS/MS | 4 Fragmentation MS/MS | References |
|---|---|---|---|---|---|---|---|---|---|---|
| 61 | Phenolic acid | **Neochlorogenic acid** [5-*O*-Caffeoylquinic acid] | $C_{16}H_{18}O_9$ | 354.3087 | 353 | | 173; 111 | | | Andean blueberry [28]; *Stevia rebaudiana* [29]; *Rosa rugosa* [30]; *Lonicera japonicum* [34]; *Euphorbia hirta* [59]; *Crataegus monogyna, Sambucus nigra* [67] |
| 62 | Phenolic acid | **Rosmarinic acid** | $C_{18}H_{16}O_8$ | 360.3148 | | 361 | 343; 327; 301; 253; 19; 161 | 253; 121 | 225; 210; 179 | *Rosmarinus officinalis* [20]; *Mentha* [26]; *Rosa rugosa* [30]; *Mentha* [70]; Huolisu Oral Liquid [71]; Rosemary [72] |
| 63 | Phenolic acid | **5-Hydroxy feruloyl hexose** * | $C_{16}H_{20}O_{10}$ | 372.3240 | | 373 | 211; 277; 354 | 175 | | millet grains [23] |
| 64 | Phenolic acid | **Salvianolic acid D** * | $C_{20}H_{18}O_{10}$ | 418.3509 | 417 | | 373 | 347; 303 | | *Mentha* [70,73]; *Salvia multiorrizae* [74] |
| 65 | Phenolic acid | **Salvianolic acid B** [Danfensuan B] * | $C_{36}H_{30}O_{16}$ | 718.6138 | | 719 | 521; 199 | 475 | | *Bougainvillea* [21]; *Mentha* [50]; Huolisu Oral Liquid [71]; *Mentha* [73]; *Salvia miltiorrhiza* [74] |
| 66 | Stilbene | **Pinosylvin** [3,5-Stilbenediol; Trans-3,5-Dihydroxystilbene] * | $C_{14}H_{12}O_2$ | 212.2439 | | 213 | 195; 171 | 143 | 127 | *Pinus resinosa* [75]; *Pinus sylvestris* [76] |
| 67 | Stilbene | **Resveratrol** [trans-Resveratrol; 3,4′,5-Trihydroxystilbene; Stilbentriol] * | $C_{14}H_{12}O_3$ | 228.2433 | | 229 | 169; 210; 141; 115 | 141 | 113 | *A. cordifolia; F. glaucescens; F. herrerae* [25]; *Radix polygoni multiflori* [42]; *Embelia* [56]; *Vine stilbenoids* [77] |
| 68 | Stilbene | **3-Hydroxyresveratrol** [Piceatannol] * | $C_{14}H_{12}O_4$ | 244.2427 | | 245 | 199; 112 | 112 | | *G. linguiforme* [25]; Vine stilbenoids [77]; *Oenocarpus bataua* [78] |
| 69 | Lignan | **Pinoresinol** * | $C_{20}H_{22}O_6$ | 358.3851 | | 359 | 340; 208 | 322; 196 | 274; 214 | *Passiflora incarnata* [51]; *Punica granatum* [65]; *Eucommia cortex* [79]; Lignans [80] |
| 70 | Lignan | **Arctigenin** * | $C_{21}H_{24}O_6$ | 372.4117 | | 373 | 354; 336; 283; 252; 211 | 336; 318; 288; 252; 218 | 288; 236; 197 | Lignans [80]; *Triticum aestivum* [81]; *Forsythia* [82] |
| 71 | Coumarin | **3,4,5-Trimethoxycoumarin** * | $C_{12}H_{12}O_5$ | 236.2207 | | 237 | 192; 206; 178 | 132 | 130; 117 | Propolis [31] |
| 72 | Coumarin | **Fraxin** (Fraxetin-8-*O*-glucoside) * | $C_{16}H_{18}O_{10}$ | 370.3081 | | 371 | 191 | 127 | | *Vitis vinifera* [45]; *Actinidia* [69]; *Solanum tuberosum* [83] |
| 73 | Anthocyanidin | **Anthocyanidin** [cyanidin chloride; Cyanidin] * | $C_{15}H_{11}O_{6+}$ | 287.2442 | | 287 | 213; 195; 167 | 196; 163; 125 | | *F. herrerae* [25]; Andean blueberry [28]; *Malpighia emarginata* [84] |
| 74 | Anthocyanidin | **Petunidin** * | $C_{16}H_{13}O_{7+}$ | 317.2702 | | 318 | 256; 300 | 228; 212; 184 | 212 | *A. cordifolia; C. edulis* [25] |
| 75 | Anthocyanidin | **Cyanidin-3-*O*-glucoside** [Cyanidin 3-*O*-beta-D-Glucoside; Kuromarin] * | $C_{21}H_{21}O_{11+}$ | 449.3848 | 447 | | 285; 195 | 255 | | *Triticum aestivum* [22]; *Malpighia emarginata* [84] |
| 76 | Anthocyanidin | **Delphinidin *O*-pentoside** * | $C_{20}H_{19}O_{11}$ | 435.3583 | | 435 | 303; 245 | 245; 149 | | Andean blueberry [28]; Myrtle [41]; *Gaultheria mucronata; Gaultheria antarctica* [85] |
| 77 | Anthocyanidin | **Pelargonidin 3-*O*-(6-*O*-malonyl-beta-D-glucoside)** * | $C_{24}H_{23}O_{13}$ | 519.4388 | | 519 | 271 | 253 | | *Gentiana lutea* [86]; Wheat [87] |
| 78 | Anthocyanidin | **Cyanidin 3-(6″-Succinyl-Glucoside)** [Cyanidin 3-(6″-*O*-succinoyl-Beta-D-Glucopyranoside)] * | $C_{25}H_{25}O_{14}$ | 549.4576 | | 549 | 286 | 268 | 240 | Wheat [87] |
| 79 | Anthocyanidin | **Delphinidin malonyl hexoside** * | $C_{24}H_{23}O_{15}$ | 551.4304 | | 551 | 465; 425; 287; 198 | 271; 157 | | *F. glaucescens* [25] |
| 80 | Anthocyanidin | **Cyanidin-3-*O*-dioxayl-glucoside** * | $C_{31}H_{28}O_{12}$ | 592.5468 | | 593 | 287; 165 | 213; 153 | | *Rubus ulmifolius* [62] |
| 81 | Anthocyanidin | **Delphinidin 3,5-dihexoside** * | $C_{27}H_{31}O_{17}$ | 627.5248 | | 627 | 413; 227 | 227; 351 | | *F. herrerae* [25]; Andean blueberry [28]; *Berberis microphylla* [85] |

**Table A1.** *Cont.*

| No. | Class of Compounds | Identified Compounds | Formula | Mass | Molecular Ion [M − H]− | Molecular Ion [M + H]+ | 2 Fragmentation MS/MS | 3 Fragmentation MS/MS | 4 Fragmentation MS/MS | References |
|---|---|---|---|---|---|---|---|---|---|---|
| 82 | Tannin | Prodelphinidin A-type * | $C_{30}H_{26}O_{13}$ | 594.5286 | | 595 | 406; 287; 245 | 241; 213; 165; 153 | 213 | *Vitis vinifera* [45] |
| 83 | Hydrolysable tannin | (S)-Flavogallonic acid | $C_{21}H_{10}O_{13}$ | 470.2963 | | 471 | 407; 321; 247; 205 | 205; 307; 389 | 177; 131 | *Terminalia arjuna* [46]; *Rosa rugosa* [47] |
| 84 | Ellagitannin | Punicalin alpha * | $C_{34}H_{22}O_{22}$ | 782.5253 | | 783 | 721; 449; 599; 535 | 596 | | *Myrtle* [41]; *Terminalia arjuna* [46]; *Punica granatum* [65] |
| 85 | Phenylpropanoid (cinnamic alcohol glycoside) | Coniferin [Coniferyl Alcohol Beta-D-Glucoside] * | $C_{16}H_{22}O_8$ | 342.3411 | | 343 | 240 | 183 | 127 | *Hedyotis diffusa* [27]; *Rhodiola crenulata* [88] |
| 86 | Gallate ester | Ethyl gallate * | $C_9H_{10}O_5$ | 198.1727 | 197 | | 169; 125 | 124 | | *Bougainvillea* [21]; *Terminalia arjuna* [46]; *Euphorbia hirta* [59] |
| 87 | Gallate ester | Beta-Glucogallin [1-O-Galloyl-Beta-D-Glucose; Galloyl glucose; Monogalloyl glucose] * | $C_{13}H_{16}O_{10}$ | 332.2601 | | 333 | 273; 227; 169 | 169; 191; 209 | | *Syzygium aromaticum* [18]; *Terminalia arjuna* [46]; *Euphorbia hirta* [59]; *Cassia granidis* [63] |
| 88 | Dihydrochalcone | Phloretin [Dihydronaringenin; Phloretol] * | $C_{15}H_{14}O_5$ | 274.2687 | | 275 | 257; 229; 215 | 255; 239; 229; 210 | | *G. linguiforme* [25]; *Punica granatum* [65]; *Malus toringoides* [89] |
| 89 | Flavonoid | Diphylloside B * | $C_{38}H_{48}O_{19}$ | 808.7763 | | 809 | 647; 592; 531; 483; 431; 369; 317 | 533; 484; 419; 369; 269 | 419 | Huolisu Oral Liquid [71] |
| 90 | Flavonoid | Demethylanhydroicaritin-7-O-glucopyranosyl-3-O-acetylated rhamnopyranosyl-xylopyranoside * | $C_{39}H_{48}O_{20}$ | 836.7854 | | 837 | 675; 603; 541; 503; 403 | 441; 341 | 341; 241 | Huolisu Oral Liquid [71] |
| | **OTHERS** | | | | | | | | | |
| 91 | Cyclohexenecarboxylic acid | Shikimic acid [L-Schikimic acid] * | $C_7H_{10}O_5$ | 174.1513 | 173 | | 111 | | | *A. cordifolia* [25]; *Camellia kucha* [38]; *Euphorbia hirta* [59] |
| 92 | Vitamin | L-Ascorbic acid [Vitamin C] | $C_6H_8O_6$ | 176.1241 | 175 | | 127 | | | *Potato leaves* [35]; *Strawberry, Lemon, Papaya* [39] |
| 93 | Monoterpenoid | Methyl eugenol * | $C_{11}H_{14}O_2$ | 178.2277 | | 179 | 161 | 133 | | *Ocimum* [19]; *Olive leaves* [90] |
| 94 | Omega-hydroxy amino acid | Hydroxy decenoic acid * | $C_{10}H_{18}O_3$ | 186.2481 | | 187 | 169; 142 | 141 | | *F. glaucescens* [25] |
| 95 | Essential amino acid | L-Tryptophan [Tryptophan; (S)-Tryptophan] * | $C_{11}H_{12}N_2O_2$ | 204.2252 | | 205 | 186; 158 | 146; 169 | 144; 118 | Rapeseed petals [36]; *Camellia kucha* [38]; *Passiflora incarnata* [51]; *Euphorbia hirta* [59]; Huolisu Oral Liquid [71] |
| 96 | Sesquiterpenoid | Caryophyllene oxide [Caryophyllene-alpha-oxide] * | $C_{15}H_{24}O$ | 220.3505 | | 221 | 161 | 147 | | *Olive leaves* [90] |
| 97 | | 3,4,5-Trimethoxyphenylacetic acid | $C_{11}H_{14}O_5$ | 226.2259 | | 227 | 127; 145; 169; 199 | 145; 117 | 127 | *Rosa rugosa* [30] |
| 98 | Omega-5 fatty acid | Myristoleic acid [Cis-9-Tetradecanoic acid] * | $C_{14}H_{26}O_2$ | 226.3550 | | 227 | 209; 127 | 139 | | *F. glaucescens* [25] |
| 99 | Quaianolide sesquiterpene lactone | Dehydrocostus Lactone * | $C_{15}H_{18}O_2$ | 230.3022 | | 231 | 214 | 168 | | Weichang'an Pill [91] |
| 100 | Germacranolide | Costunolide * | $C_{15}H_{20}O_2$ | 232.3181 | | 233 | 186 | 168; 131 | 155 | Weichang'an Pill [91] |
| 101 | Biphenyl derivative | Randaiol * | $C_{15}H_{14}O_3$ | 242.2699 | | 243 | 225; 211; 182 | 182; 167; 132 | 166 | *Magnolia officinalis* [92] |
| 102 | Peptide | 5-Oxo-L-propyl-L-isoleucine * | $C_{11}H_{18}N_2O_4$ | 242.2716 | | 243 | 197 | 165 | 137 | *Potato leaves* [35] |
| 103 | Hydroxy monocarboxylic acid | Hydroxy myristic acid [2S-Hydroxytetradecanoic acid; Alpha-Hydroxy Myristic acid] * | $C_{14}H_{28}O_3$ | 244.3703 | | 245 | 229; 222; 211; 201 | 227; 211; 201 | | *F. pottsii* [25] |

**Table A1.** *Cont.*

| No. | Class of Compounds | Identified Compounds | Formula | Mass | Molecular Ion [M − H]⁻ | Molecular Ion [M + H]⁺ | 2 Fragmentation MS/MS | 3 Fragmentation MS/MS | 4 Fragmentation MS/MS | References |
|---|---|---|---|---|---|---|---|---|---|---|
| 104 | Medium-chain fatty acid | Hydroxy dodecanoic acid * | $C_{12}H_{22}O_5$ | 246.3001 | | 247 | 229; 202; 174; 156 | 183; 156; 144 | 156 | *F. glaucescens* [25] |
| 105 | Acyclic alcohol nitrile glycoside | Rhodiocyanoside A * | $C_{11}H_{17}NO_6$ | 259.2558 | | 260 | 186; 232 | 168 | 141 | *Rhodiola rosea* [93]; *Rhodiola sacra* [94] |
| 106 | Naphthoquinone | Spinochrome A * | $C_{12}H_8O_7$ | 264.1877 | | 265 | 247 | 219 | | *Rhus coriaria* [32] |
| 107 | Aporphine alkaloid | Anonaine * | $C_{17}H_{15}NO_2$ | 265.3065 | | 266 | 247; 190; 166 | 166 | | *Magnolia officinalis* [92] |
| 108 | Ribonucleoside composite of adenine (purine) | Adenosine * | $C_{10}H_{13}N_5O_4$ | 267.2413 | | 268 | 136 | | | *Lonicera japonica* [34]; Huolisu Oral Liquid [71] |
| 109 | | 3,4,8,9,10-Penthahydroxydibenzo [b,d]pyran-6-one * | $C_{13}H_8O_7$ | 276.1984 | | 277 | 175; 231; 259 | 131; 177 | | *Terminalia arjuna* [46] |
| 110 | | Linoleic acid amide * | $C_{18}H_{33}NO$ | 279.4607 | | 280 | 262 | 244; 196; 164; 128 | 226; 196; 164 | Propolis [31]; *Rhus coriaria* [32] |
| 111 | | Oleamide * | $C_{18}H_{35}NO$ | 281.4766 | | 282 | 247 | 173; 201; 145 | 145 | Propolis [31] |
| 112 | Terpenoid | Rugosic acid A | $C_{15}H_{22}O_5$ | 282.3322 | | 283 | 239; 265; 167 | 211 | 193; 170 | *Rosa rugosa* [95] |
| 113 | Alkaloid | Mesembrenol * | $C_{17}H_{23}NO_3$ | 289.3694 | | 290 | 272; 146 | 224; 182 | 164 | *Sceletium* [96] |
| 114 | Alkaloid | Mesembranol * | $C_{17}H_{25}NO_3$ | 291.3853 | | 292 | 274; 226; 111 | 121 | | *A. cordifolia* [25]; *Sceletium* [96] |
| 115 | | Brevifolincarboxylic acid * | $C_{13}H_8O_8$ | 292.4131 | 291 | | 247 | 219; 203; 191; 175; 147 | 191 | *Euphorbia hirta* [59] |
| 116 | Alkaloid | 3′-Methoxy-4′-O-methyl joubertimine * | $C_{18}H_{25}NO_3$ | 303.3960 | | 304 | 257; 195; 153 | 231; 149 | 213 | *A. cordifolia* [25] |
| 117 | Diterpenoid | Tanshinone IIB [(S)-6-(Hydroxymethyl)-1,6-Dimethyl-6,7,8,9-Tetrahydrophenanthro [1,2-B]Furan-10,11-Dione] * | $C_{19}H_{18}O_4$ | 310.3438 | | 311 | 293; 265; 253; 228; 181 | 264; 192; 159 | | Huolisu Oral Liquid [72]; *Salviae Miltiorrhizae* [97] |
| 118 | Oxylipins | 11-Hydroperoxy-octadecatrienoic acid * | $C_{18}H_{30}O_4$ | 310.4284 | 309 | | 291; 247; 198; 183 | 181 | | Potato leaves [35] |
| 119 | Tyramines | N-Feruloyl tyramine * | $C_{18}H_{19}NO_4$ | 313.3478 | | 314 | 296; 236; 175 | 222; 206; 178 | 222; 194; 180 | *Bougainvillea* [21] |
| 120 | Terpenoid trilactone | Bilobalide [(-)-Bilobalide] * | $C_{15}H_{18}O_8$ | 326.2986 | 325 | | 183 | 119; 199 | | *Malus toringoides* [89]; *Ginkgo biloba* [98,99] |
| 121 | Oxylipins | 9,10-Dihydroxy-8-oxooctadec-12-enoic acid [oxo-DHODE; oxo-Dihydroxy-octadecenoic acid] * | $C_{18}H_{32}O_5$ | 328.4437 | 327 | | 229; 291 | 211; 125 | 183 | *Phyllostachys nigra* [24]; *Bituminaria bituminosa* [100] |
| 122 | Oxylipins | 13- Trihydroxy-Octadecenoic acid [THODE] * | $C_{18}H_{34}O_5$ | 330.4596 | 329 | | 291; 309; 239; 211; 197; 171 | 273; 217; 179 | 255; 228 | *Sasa veitchii* [24]; *Bituminaria bituminosa* [100]; Broccoli [101] |
| 123 | Sceletium alkaloid | O-acetyl mesembrenol * | $C_{19}H_{25}NO_4$ | 331.4061 | 330 | | 270; 226; 198 | 226; 209; 166 | 166 | *A. cordifolia* [25] |
| 124 | Diterpenoid | Carnosic acid | $C_{20}H_{28}O_4$ | 332.4339 | 331 | | 287; 243; 187 | 259 | 215 | *Rosmarinus officinalis* [20]; Rosemary [72]; *Lepechinia* [102] |
| 125 | | Dihydroxy eicosatrienoic acid * | $C_{20}H_{34}O_4$ | 338.4816 | | 339 | 321; 177 | 303; 274; 233 | 178; 148 | *G. linguiforme*; *A. cordifolia*; *C. edulis* [25] |
| 126 | Berberine alkaloid | Palmatine [Berbericinine; Burasaine] * | $C_{21}H_{22}NO_4$ | 352.4037 | | 353 | 335; 308; 270; 235; 195 | 317; 243; 215; 160 | | *Ocotea* [103]; Palmatine [104] |
| 127 | Unsaturated fatty acid | Dihydroxy docosanoic acid * | $C_{22}H_{44}O_4$ | 372.5824 | | 373 | 341 | 327 | | *A. cordifolia*; *F. pottsii* [25] |
| 128 | Unsaturated fatty acid | Pentacosenoic acid * | $C_{25}H_{48}O_2$ | 380.6474 | | 381 | 363; 334; 290; 261; 231 | 342; 303; 276 | | *F. glaucescens* [25] |
| 129 | Sterol | Campesterol [Dihydrobrassicasterol] * | $C_{28}H_{48}O$ | 400.6801 | | 401 | 383; 369; 337; 310; 279 | 350; 321; 285; 249 | 262 | *A. cordifolia*; *C. edulis* [25]; *Oryza sativa* [105] |
| 130 | Alkaloid | Erysothiopine * | $C_{19}H_{21}NO_7S$ | 407.4375 | | 408 | 389 | 345; 183 | 299; 161 | *Camellia kucha* [38] |

**Table A1.** *Cont.*

| No. | Class of Compounds | Identified Compounds | Formula | Mass | Molecular Ion [M − H]⁻ | Molecular Ion [M + H]⁺ | 2 Fragmentation MS/MS | 3 Fragmentation MS/MS | 4 Fragmentation MS/MS | References |
|---|---|---|---|---|---|---|---|---|---|---|
| 131 | Sterol | **Stigmasterol** [Stigmasterin; Beta-Stigmasterol] * | $C_{29}H_{48}O$ | 412.6908 | | 413 | 301 | 188 | | *A. cordifolia; F. pottsii* [25]; *Hedyotis diffusa* [27]; Olive leaves [90] |
| 132 | Iridoid monoterpenoid | **Dihydroisovaltrate** * | $C_{22}H_{32}O_8$ | 424.4847 | | 425 | 365; 281 | 309; 235 | | *Rhus coriaria* [32] |
| 133 | Anabolic steroid; Androgen; Androgen ester | **Vebonol** * | $C_{30}H_{44}O_3$ | 452.6686 | | 453 | 435; 336; 226 | 336 | 209 | *Rhus coriaria* [32]; *Hylocereus polyrhizus* [106] |
| 134 | Triterpenoid | **Betulonic acid** [Betunolic acid; Liquidambaric acid] * | $C_{30}H_{46}O_3$ | 454.6844 | | 455 | 437; 357; 245 | 176; 395; 336; 261; 213 | | *Rhus coriaria* [32] |
| 135 | Triterpenoid | **Pomolic acid** * | $C_{30}H_{48}O_4$ | 472.6997 | | 473 | 413; 214 | 395; 255 | 241 | *Sanguisorba officinalis* [68]; *Malus domestica* [107] |
| 136 | Thromboxane receptor antagonist | **Vapiprost** * | $C_{30}H_{39}NO_4$ | 477.6350 | | 478 | 337; 263; 218; 173 | 181; 128 | | *Rhus coriaria* [32]; *Hylocereus polyrhizus* [106] |
| 137 | Ursane triterpene | **Annurcoic acid** * | $C_{30}H_{46}O_5$ | 486.6922 | 485 | | 467; 423 | 424; 393; 335 | 413 | Annurca apple [108] |
| 138 | Pentacyclic triterpenoid | **Methyl arjunolate** * | $C_{31}H_{50}O_5$ | 502.7257 | | 503 | 485; 205 | 397; 197 | | *G. linguiforme; C. edulis* [25] |
| 139 | Indole sesquiterpene alkaloid | **Sespendole** * | $C_{33}H_{45}NO_4$ | 519.7147 | | 520 | 185; 502 | 125 | | *Rhus coriaria* [32]; *Hylocereus polyrhizus* [106] |
| 140 | Schisandrin | **Benzoylgomisin H** * | $C_{30}H_{34}O_8$ | 522.5862 | | 523 | 504; 448; 399; 369 | 486; 447; 424; 405; 362 | 424; 350; 290; 252 | *Schisandra chinensis* [109,110] |
| 141 | Carotenoid | **(all-E)-alpha-Cryptoxanthin** | $C_{40}H_{56}O$ | 552.872 | | 553 | 535 | 517; 499; 443; 395 | 499; 457; 363; 307 | *Carica papaya* [111]; *Physalis peruviana* [112]; *Rosa rugosa* [113] |
| 142 | | **N',N',N'''- Tri-p-coumaroyl spermidine** | $C_{34}H_{37}N_3O_6$ | 583.6741 | | 584 | 565; 467; 438; 387; 335 | 204; 292; 218; 147 | 147 | *Rosa rugosa* [11]; Propolis [31] |
| 143 | | **N',N',N'''- Di-p-coumaroyl caffeoyl spermidine** | $C_{34}H_{37}N_3O_7$ | 599.6735 | | 600 | 582; 497; 438; 420 | 419; 328; 292; 274 | 147 | *Rosa rugosa* [11] |
| 144 | Cycloartanol [Steroids] | **Cyclopassifloic acid glucoside** * | $C_{37}H_{62}O_{12}$ | 698.8810 | | 699 | 537; 421; 365 | 520 | | *Passiflora incarnata* [51] |
| 145 | Carotenoid | **(all-E)-violaxanthin caproate** * | | 755 | | 755 | 719; 645; 566; 425 | 657; 620 | | Carotenoids [114] |
| 146 | Derivative of Chlorophylle | **Pheophytin b** * | $C_{55}H_{72}N_4O_6$ | 885.1834 | | 886 | 607 | 547 | 475; 419 | *Physalis peruviana* [112,115] |

* Compounds identified for the first time in genus *Rosa*.

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
