# Peer review of "Rosa davurica Pall., Rosa rugosa Thumb., and Rosa acicularis Lindl. Originating from Far Eastern Russia: Screening of 146 Chemical Constituents in Three Species of the Genus Rosa"

_applsci, doi:10.3390/app12199401_

Round 1

Reviewer 1 Report

Dear Authors,

The received manuscript entitled "Rosa davurica Pall., Rosa rugosa Thunb., and Rosa acicularic Lindl. originating from Trans-Baikal Region, and Far East: A High-Resolution Mass Spectrometric Approach for the Comprehensive Characterization of Phytochemicals" submitted to Applied Biosciences and Bioengineering section in Applied Sciences journal presents bioactive compounds in three members of the genus Rosa identified with the aid of liquid chromatography combined with ion trap. The novelty of the work is undoubtedly the 115 compounds that were identified for the first time in the genus Rosa. The described method (HPLC- ion trap) is well presented and wisely chosen however I would recommend continuing this interesting work with HPLC- MS/MS to double check the bioactive compounds found in the samples. It is often done like that- first screening and then continuing work with precise instrumentation for target analytes. 

I also feel that the Conclusion section should be improved. It is good to read that the Authors are aware that this is preliminary study and therefore future research will be intensified. 

Author Response

Dear Reviewer.

Our team of authors is extremely grateful to you for your help in pointing out the errors and the overall work on the article. We tried to make a corrections in the text as much as possible and revised inaccuracies and incorrectness's.

We also completely rewrote the Conclusions.

Much obliged to you,

Sincerely yours,

Dr. Razgonova Mayya

Reviewer 2 Report

General Comments: 

Comment 1: The key points should be focused on nicely in the 'Introduction' section. The authors need to change the introduction section considerably. Try to include the existing research limitations and also how the present research unravels those limits.

Comment 2: Few grammatical errors are observed throughout the article, which needs to be corrected.

Comment 3: The results are clearly presented and adequately addressed.

Comment 4: The discussion is well-written and adequately addressed.

Comment 5: The author should provide critical points, the current study's contribution to literature, and what messages are provided with the present study in the conclusion.

Specific Comments: 

-Figures 1 and 2 should be combined into one figure. 

 -Title is very misleading, pls revisit and change it for better understanding.

-Study background must be improved; You may take consultation with the articles and cite them for strong evidence. https://doi.org/10.1016/j.biopha.2022.112932;  https://doi.org/10.1016/j.phrs.2022.106398 ; https://doi.org/10.3390/molecules27144374; https://doi.org/10.1016/j.jep.2021.114588

Author Response

Dear Reviewer.

Our team of authors is extremely grateful to you for your help in pointing out the errors and the overall work on the article. We tried to make a corrections in the text as much as possible and revised inaccuracies and incorrectness's.

We have changed the title of the article, also included links to your respected articles about bioactive substances and polyphenol groups.

We also completely rewrote the Conclusions.

Much obliged to you,

Sincerely yours,

Dr. Razgonova Mayya
